# The Impact of Exposure to Hexavalent Chromium on the Incidence and Mortality of Oral and Gastrointestinal Cancers and Benign Diseases: A Systematic Review of Observational Studies, Reviews and Meta-Analyses

Konstantinos Katsas [1,2,*], Dimitrios V. Diamantis [1], Athena Linos [1], Theodora Psaltopoulou [3] and Konstantinos Triantafyllou [4]

[1] Civil Law Non-Profit Organization of Preventive Environmental and Occupational Medicine, PROLEPSIS Institute, 7 Fragoklisias Street, 15121 Athens, Greece; dimitrios.v.diamantis@gmail.com (D.V.D.); a.linos@prolepsis.gr (A.L.)

[2] Medical School, National and Kapodistrian University of Athens, 75 Mikras Asias Street, 11527 Athens, Greece

[3] Department of Hygiene, Epidemiology and Medical Statistics, School of Medicine, University of Athens, 75 Mikras Asias Street, 11527 Athens, Greece; tpsaltop@hotmail.com

[4] Hepatogastroenterology Unit, Second Department of Propaedeutic Internal Medicine, Medical School, National and Kapodistrian University of Athens, Attikon University General Hospital, 1 Rimini Street, 12462 Athens, Greece; ktriant@med.uoa.gr

[*] Correspondence: katkonstantinos@gmail.com

**Abstract:** Background: Limited evidence suggests a possible connection between natural or occupational exposure to chromium and an increased risk of gastrointestinal cancer. The main objective of this study is to investigate the literature regarding chromium exposure and gastrointestinal health issues (i.e., cancer). Methods: A systematic literature search was performed using PubMed, Google Scholar and ScienceDirect. Included observational studies were assessed for their risk of bias. Results: 16 observational studies and 7 reviews and meta-analyses met the inclusion criteria. Most of the studies investigated gastric and hepatocellular cancer, followed by colorectal, oral, esophageal and pancreatic cancer. There is a limited amount of evidence regarding non-malignant gastrointestinal diseases. Chromium exposure is suspected to increase gastric and colorectal cancer risks. We did not find any convincing indications for increases in oral, esophageal and hepatocellular cancer. Pancreatic, gallbladder and extrahepatic bile ducts carcinogenesis is likely not associated with chromium exposure. Conclusion: We found weak evidence that chromium exposure is associated with gastric and colorectal cancer. Our review also highlights the existing controversial evidence regarding oral, esophageal and hepatocellular cancer, as well as the gap in studies investigating small intestinal cancer and non-malignant gastrointestinal health issues.

**Keywords:** hexavalent chromium; gastrointestinal cancer; mortality; incidence; systematic review

## 1. Introduction

### 1.1. Rationale

Chromium (Cr) is a steel gray, lustrous, hard metal extracted from chromite ores. It is naturally located in rocks, animals, plants, soil and volcanic dust and gases in various forms, such as trivalent chromium, an essential micronutrient for humans, and hexavalent chromium (Cr(VI)), a common industry product [1]. Cr(VI) compounds are groups of chemicals that are valued for their different properties (i.e., corrosion-resistance, durability and hardness) and are used widely in stainless steel (ferrochrome alloy) and chrome plating [2]. Cr(VI) in the workplace can be found in chromate compounds (barium, calcium, lead, potassium, silver, sodium, ammonium and zinc), or even in the form of lead chromate

oxide. These compounds can be located in batteries, construction and building materials (i.e., for flooring, tile, sinks, bathtubs, mirrors, etc.), repair adhesives (glues), paint/stain- and related products and surface sealers [3,4].

Exposure to Cr(VI) may either be natural (water or topsoil with high concentrations levels of chromium) or occupational, occurring primarily among metal and chemical manufacturing workers. According to the Occupational Safety and Health Administration [3], workers with a higher risk of Cr(VI) exposure are those occupied in chromate production, stainless steel welding, chrome plating/electroplating, chrome pigment production, ferrochrome production industries and leather tanning. Professions involving painting, abrasive blasting, copying machine and printer toner powder maintenance and disposal, concreting, welding, cutting, brazing, soldering, torching, and the manufacturing of batteries, candles, dyes and rubber-based products are also considered as high risk.

The adverse effects of Cr(VI) exposure have been a research interest of various international agencies [2,5–7]. Many suggest that occupational exposure to Cr(VI) may lead to chronic respiratory diseases, including asthma, allergic dermatitis and lung cancer [2,5]. According to the National Institute for Occupational Safety and Health (NIOSH), some studies have shown evidence of significant impacts on cancer risks, such as oral, hepatocellular, esophagus and overall cancer risk. However, the overall pattern of augmented risk is inconsistent [2]. In a recent review, the authors concluded that occupational exposure to Cr(VI) may increase the risk of lung, nose and nasal sinus cancer, while outcomes related to gastric and laryngeal cancer are only suggestive [8]. The existing evidence from human studies investigating the risk for gastrointestinal cancer due to Cr(VI) exposure is limited, inconclusive and often conflicting. An even smaller body of literature is also available for non-malignant gastrointestinal diseases. Some studies suggest that Cr(VI) exposure may contribute to some adverse gastrointestinal effects, such as abdominal pain, duodenal ulcers, gastritis, gastric cramps, severe liver damage and cirrhosis, especially in workers in the chrome plating industry [9].

### 1.2. Objectives

The lack of consensus regarding the relationship between gastrointestinal health issues such as cancer and Cr(VI) exposure has led us to investigate the current and past literature to shed light on this complex relationship. The primary objectives of our study were: (1) to collect all the established evidence through a systematic literature review of the published reviews and meta-analyses in the last decade and (2) to update the existing knowledge, conducting a systematic literature review of all observational studies published during the same period, regarding the impact of Cr(VI) exposure on gastrointestinal cancer. As a secondary outcome, we evaluated the effects of Cr(VI) exposure on benign gastrointestinal diseases using the same methodological approach.

## 2. Materials and Methods

### 2.1. Registration

This study was conducted following the 2020 Preferred Reporting Items for Systematic Review and Meta-Analysis (PRISMA) [10]. The systematic review protocol was registered in the International Prospective Register of Systematic Reviews (PROSPERO) under registration ID: CRD42023409604.

### 2.2. Eligibility Criteria

Observational human studies, reviews and meta-analyses published from January 2012 to January 2023 were included. A study was considered eligible if the study's sample (humans) included participants who were likely to be exposed to Cr(VI). This translates to studies with environmental exposure (non-occupational) or occupational exposure to Cr(VI). The Occupational Safety and Health Administration report [3] was used to identify groups of participants with an increased risk of occupational exposure to Cr(VI), namely workers in leather tanning, stainless steel welding, metal plating, chrome pigment produc-

tion, chromate production, chrome plating/electroplating, abrasive blasting, ferrochrome production industries, printing industries, rubber manufacturing industries and cement workers or dye makers. Studies in which the authors mentioned that the workers were exposed to Cr(VI) were also included.

Studies were excluded from the review if any of the following criteria were met: (a) the retrieved article was a case report, case series, conference abstract or expert opinion, (b) studies involving subjects under 18 years of age, (c) studies with a relatively small sample ($n < 50$), (d) ecological studies not mentioning exposure to Cr(VI), (e) studies not published in English. If more than one study was published with the same data, only the most recent results were included.

### 2.3. Information Sources and Search Strategy

A thorough systematic literature search was conducted using the databases PubMed, Google Scholar and ScienceDirect to collect all epidemiological data from January 2012 to January 2023. The following combinations of keywords was used to gather all relevant publications:

- Exposure assessment: leather tanning, stainless steel, cement, welding, metal/chrome plating, chrome pigment, chromate production, electroplating, ferrochrome, abrasive blasting, battery/candle/dye/rubber maker, printers, brazing, soldering, Cr(VI), chromium, heavy metal, toxic metal
- Outcome assessment: stomach–gastric/anal/bile duct/colon/esophageal/gallbladder/liver/pancreatic/rectal/small intestine/gastric cancer (or tumor), gastritis, gastroesophageal reflux disease, ulcer, irritable bowel syndrome, hemorrhoids, Crohn, ulcerative colitis, constipation, gastrointestinal bleeding, diverticulitis, celiac disease, gallstones, cholelithiasis, cirrhosis.

In more detail, regarding the outcome assessment, for the first group (gastrointestinal cancers and oral cancer), all studies were included with at least one type of gastrointestinal cancer (esophageal cancer, gastric cancer, small intestinal cancer, colon cancer, rectal cancer, hepatocellular cancer, pancreatic cancer, gallbladder and extrahepatic bile ducts cancer) or oral cancer. The second group (benign or non-malignant gastrointestinal diseases) included all non-cancer gastrointestinal diseases, such as gastritis, gastroesophageal reflux disease, ulcer, irritable bowel syndrome, hemorrhoids, Crohn's disease, ulcerative colitis, constipation, gastrointestinal bleeding, diverticulitis, celiac disease, gallstones, cholelithiasis and cirrhosis.

Two separate literature searches were conducted using the referred keywords, the first excluding all reviews, systematic reviews and meta-analyses, and the second only with the three mentioned types of studies. The exact syntax is presented in Supplementary Table S1.

### 2.4. Study Selection

Two independent reviewers (KK, DVD) conducted the study selection process to mitigate inclusion bias. After removing the duplicates, the reviewers performed title and abstract screening for all the studies and determined whether each study should be included. The included studies had their full text retrieved and were assessed as meeting the inclusion criteria. Both reviewers stated the reasons for study exclusion, and if a disagreement arose, a third reviewer (AL) was asked to make the final decision. The references of all the retrieved full-text articles were hand searched for relevant articles that were not included in the initial search strategy results. The study selection process results can be found in Figure 1.

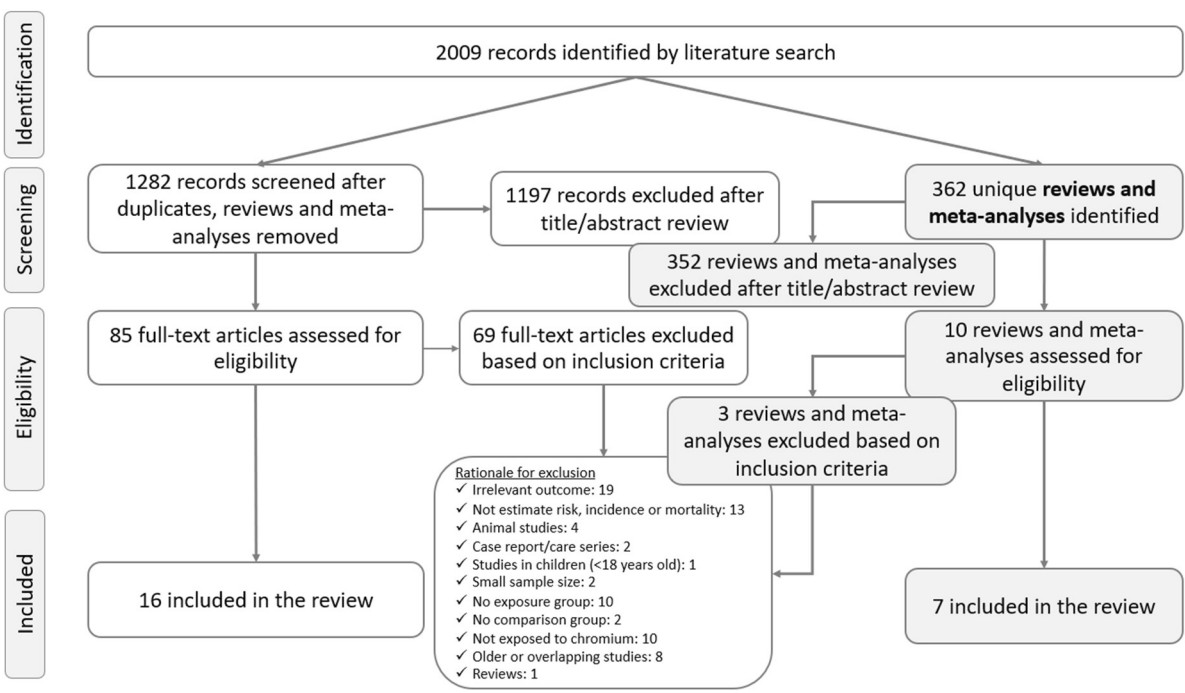

**Figure 1.** Literature search results (in accordance to PRISMA reporting).

*2.5. Data Extraction*

Table 1 presents the primary characteristics of each human observational study (study design, country of the study population, number of participants, exposure group/cases, comparison/control group, primary outcome (incidence, mortality or both), outcome measure (relative risk (RR), standardized mortality ratio (SMR), standardized incidence ratio (SIR), odds ratio (OR), hazard ratio (HR), age-standardized incidence rate (ASR)) and main results/conclusion regarding diseases of interest). The study design, period covered, number of studies included, primary outcomes and key findings for the included reviews and meta-analyses are displayed in Table 2. Table 3 describes all the types of gastrointestinal cancers and benign gastrointestinal diseases that were investigated in each study separately.

**Table 1.** Characteristics of included observational human studies.

| Author | Study Design | Study Population | Number of Study Participants | Exposure/ Target Group | Comparison Group | Outcome | Measure | Key Findings | Tier |
|---|---|---|---|---|---|---|---|---|---|
| DeBono et al., 2020 [11] | Retrospective cohort | Canada | 2.18 million workers; 81,127 workers in the exposure group | Plastics and rubber manufacturing | Rest of the workers | Incidence | HR | ↑ HR for esophageal and gastric Ca in job-specific subgroups | 2 |
| Salerno and Cucciniello, 2019 [12] | Retrospective cohort | Italy | 899 workers | Electroplating factory workers | Regional population | Mortality | SMR | ↑ SMR for digestive tract and hepatocellular Ca | 2 |
| Sciannameo et al., 2019 [13] | Retrospective cohort | Italy | 2991 workers | Electroplating factory workers | Non-exposed workers | Mortality | HR | NS different risk for GI Ca with exposure to Cr | 1 |
| Gibb et al., 2015 [14] | Prospective cohort | USA | 2354 workers | Chromate production | National data | Mortality | SMR | NS in SMR for all GI Ca | 2 |
| Girardi et al., 2015 [15] | Retrospective cohort | Italy | 127 workers | Chromium thin-layer plating | Northern Italy population | Mortality | SMR | ↑ SMR for pancreatic Ca | 2 |
| Gerosa et al., 2013 [16] | Retrospective cohort | Italy | 2983 workers | Electroplating workers | Northern Italy population | Mortality | SMR | ↑ SMR for rectal Ca | 2 |
| Wu et al., 2013 [17] | Retrospective cohort | Taiwan | 4962 workers | Shipbreaking Workers | National data | Mortality | SMR | (1) ↑ SMR for oral, nasopharyngeal and hepatocellular Ca and cirrhosis in male workers (2) NS in SMR for all female workers | 2 |
| Koh et al., 2013 [18] | Retrospective cohort | Korea | 1324 male workers | Cement industry workers | National data | Incidence | SIR | (1) ↑ SIR for rectal Ca in all workers (2) ↑ SIR for gastric Ca in high dust exposure group | 2 |
| Ilychova and Zaridze, 2012 [19] | Retrospective cohort | Moscow, Russia | 4525 workers | Printing industry workers | Moscow population | Mortality | SMR | NS in SMR for all GI Ca | 2 |
| Kendzia et al., 2022 [20] | Case–control | 7 European countries [+] | 644 male cases; 1959 male controls | Workers exposed to welding fumes | Non-exposed | Incidence | OR | Regular welding and lifetime exposure was associated with an increased risk of small intestinal Ca | 1 |

**Table 1.** *Cont.*

| Author | Study Design | Study Population | Number of Study Participants | Exposure/ Target Group | Comparison Group | Outcome | Measure | Key Findings | Tier |
|---|---|---|---|---|---|---|---|---|---|
| Shah et al., 2020 [21] | Pooled case–control | 8 countries [++] | 5279 GI Ca cases; 12,297 controls | Occupational exposure to Cr | Non-exposed | Incidence | OR | ↑ Odds for gastric Ca in workers exposed to Cr | 1 |
| Kaneko et al., 2020 [22] | Case–control | Japan | 40,370 Ca cases; 26,746 controls | Various manufacturing industry categories | Non-exposed | Incidence | OR | (1) ↑ Odds for colon Ca in printing industry workers (2) ↑ Odds for pancreatic and hepatocellular Ca in leather tanning, leather products and fur workers | 1 |
| Yang et al., 2013 [23] | Pooled case–control | China | 6998 workers | Occupational exposure to Cr | Non-exposed | Incidence; Mortality | SMR, OR | (1) ↑ Odds for hepatocellular Ca in male workers exposed to Cr (2) NS in SMR for hepatocellular Ca | 2 |
| Núñez et al., 2016 [24] | Ecological study | Spain | (1) Ca Mortality data from the National Statistics Institute (2) 21,187 topsoil samples | Topsoil levels of chromium | - | Mortality | RR | Higher topsoil concentration to Cr ~ upper GI tract Ca in females | 3 |
| Chen et al., 2015 [25] | Ecological study | China | (1) Records of all residents' deaths in Suzhou (2) 1683 topsoil samples | Topsoil levels of chromium | - | Mortality | RR | NS between Cr exposure with colon, gastric and hepatocellular Ca mortality rates | 3 |
| García-Pérez et al., 2015 [26] | Ecological study | Spain | Ca Mortality data from the National Statistics Institute | Production of cement, lime, plaster | Distance from industrial facility | Mortality | RR | (1) ↑ risk for colorectal Ca (2) ↑ risk for gastric Ca, only in men (3) NS risk for the rest GI Ca | 3 |

Notes. ~: significant association with an increased risk for; ↑: significantly increased; [+] Denmark, Sweden, Latvia, France, Germany, Italy, Spain; [++] Italy, Canada, China, Russia, USA, Japan, Spain, Brazil. Tiers were calculated based on an RoB assessment [27]. Tier 1 are studies with definitely or probably low RoB, Tier 3 are studies with definitely or probably high RoB and Tier 2 are studies with moderate RoB, classified as "studies that met neither the criteria of 1st nor 3rd Tiers". Abbreviations. chromium (Cr); relative risk (RR); age-standardized incidence rate (ASR); standardized mortality ratio (SMR); standardized incidence ratio (SIR); odds ratio (OR); hazard ratio (HR); cancer (Ca), gastrointestinal (GI); not statistically significant (NS); International Classification of Diseases, Revision 7 (ICD7); study risk-of-bias (RoB).

**Table 2.** Characteristics of included reviews and meta-analyses.

| Author | Study Design | Period Covered | Human Studies | Outcome | Key Findings |
|---|---|---|---|---|---|
| den Braver-Sewradj et al., 2021 [8] | Systematic review | 2012–2018 | NA * | I&M | Cr(VI) is suspected to cause gastric Ca—limited evidence from human studies. No convincing evidence that Cr(VI) can cause colorectal, esophageal and hepatocellular Ca. Insufficient evidence that Cr(VI) may cause small intestinal, oral cavity and pancreatic Ca. |
| Hessel et al., 2021 [28] | Systematic review | 2012–2018 | NA * | I&M | No convincing evidence that Cr(VI) may have GI effects in humans. |
| Suh et al., 2019 [29] | Meta-analysis | 1980–2018 | 44 | I&M | MRR = 1.08 (95%CI [0.96, 1.21]) for gastric Ca. |
| Deng et al., 2019 [30] | Meta-analysis | 1985–2016 | 47 | I&M | MSIR = 1.30 (95%CI [1.11, 1.54]) for oral Ca ($n = 16$); MSIR = 1.20 (95%CI [1.08, 1.32]) for gastric Ca ($n = 14$); MSIR = 1.05 (95%CI [1.00, 1.11]) for digestive system Ca (esophageal, gastric, pancreatic, colon, rectum, hepatobiliary system Ca; $n = 51$); NS MSIR for each Ca separately; MSMR = 0.97 (95%CI [0.92, 1.01]) for oral and digestive system Ca (esophageal, gastric, pancreatic, colon, rectum, hepatobiliary system and intestinal Ca; $n = 99$); NS MSMR for each Ca separately |
| Donato et al., 2016 [31] | Meta-analysis | 1984–2016 | 9 | I&M | MRR = 0.93 (95%CI [0.70, 1.17]) for gastric Ca; MSMR = 0.95 (95%CI [0.65, 1.26]) for gastric Ca ($n = 7$); MSIR = 0.85 (95%CI [0.59, 1.11]) for gastric Ca ($n = 4$) |
| Welling et al., 2015 [32] | Meta-analysis | 1980–2018 | 56 | I&M | MRR = 1.27 (95%CI [1.20, 1.35]) for gastric Ca; MSMR = 1.39 (95%CI [1.28, 1.51]) for gastric Ca ($n = 44$); MSIR = 1.17 (95%CI [1.09, 1.27]) for gastric Ca ($n = 30$) |
| Cohen et al., 2014 [33] | Meta-analysis | 1980–2013 | 26 | I&M | MSMR = 1.07 (95%CI [0.72, 1.59]) for gastric Ca ($n = 5$); MSIR = 1.05 (95%CI [0.66, 1.68]) for gastric Ca ($n = 4$); MSMR = 1.05 (95%CI [0.79, 1.40]) for colorectal Ca ($n = 4$); MSIR = 1.38 (95%CI [1.02, 1.88]) for colorectal Ca ($n = 3$) |

Notes. * This systematic review refers to all studies (human and non-human) examining exposure to hexavalent chromium and adverse health effects (Ca and non-Ca), not limited to GI diseases. Abbreviations. cancer (Ca); gastrointestinal (GI); meta-standardized incidence ratio (MSIR); meta-standardized mortality ratio (MSMR); meta-relative risk (MRR); not statistically significant (NS); not available (NA); incidence and mortality (I&M).

**Table 3.** All gastrointestinal issues studied in each included research paper.

| Author | Outcome | Oral Ca | Esophageal Ca | Gastric Ca | Small Intestinal Ca | Colon Ca | Rectal Ca | Hepatocellular Ca | Pancreatic Ca | Gallbladder and Extrahepatic Bile Duct Ca | Benign GI Diseases |
|---|---|---|---|---|---|---|---|---|---|---|---|
| **Cohort, case–control and cross-sectional studies** | | | | | | | | | | | |
| DeBono et al., 2020 [11] | I | | X * | X * | | | | | X | | |
| Salerno and Cucciniello, 2019 [12] | M | X | X | X | | X | X | X * | X | | |

Table 3. *Cont.*

| Author | Outcome | Oral Ca | Esophageal Ca | Gastric Ca | Small Intestinal Ca | Colon Ca | Rectal Ca | Hepatocellular Ca | Pancreatic Ca | Gallbladder and Extrahepatic Bile Duct Ca | Benign GI Diseases |
|---|---|---|---|---|---|---|---|---|---|---|---|
| Sciannameo et al., 2019 [13] | M | X | | X | | X | X | X | X | X | |
| Gibb et al., 2015 [14] | M | X | X | X | X | X | X | X | X | X | |
| Girardi et al., 2015 [15] | M | X | X | X | | X | X | X | X * | X | |
| Gerosa et al., 2013 [16] | M | X | X | X | | X | X * | X | X | X | |
| Wu et al., 2013 [17] | M | X + | X | X | | | | X + | | X | X + |
| Koh et al., 2013 [18] | I | X | X | X + | | X | X + | X | | X | |
| Ilychova and Zaridze, 2012 [19] | M | X | X | X | | X | X | X | X | | |
| Kendzia et al., 2022 [20] | I | | | | X * | | | | | X | |
| Shah et al., 2020 [21] | I | | | X * | | | | | | | |
| Kaneko et al., 2020 [22] | I | | X | X | | X * | | X * | X * | | |
| Yang et al., 2013 [23] | Both | | | | | | | X + | | | |
| Núñez et al., 2016 [24] | M | X | X ++ | X ++ | | X | X | X | X | | |
| Chen et al., 2015 [25] | M | | | X | | X | X | X | | X | |
| García-Pérez et al., 2015 [26] | M | X | X | X + | X | X * | X * | X | X | X | |
| **Reviews and meta-analyses** | | | | | | | | | | | |
| den Braver-Sewradj et al., 2021 [8] | Both | X | X | X | X | X | X | X | X | | |
| Hessel et al., 2021 [28] | Both | | | | | | | | | | X |
| Suh et al., 2019 [29] | Both | | | X | | | | | | | |
| Deng et al., 2019 [30] | Both | X * | X | X * | X | X | X | X | X | X | |
| Donato et al., 2016 [31] | Both | | | X | | | | | | | |
| Welling et al., 2015 [32] | Both | | | X * | | | | | | | |
| Cohen et al., 2014 [33] | Both | | | X | | X * | X * | | | | |

Notes. * significant results for the total group; + significant results for males; ++ significant results for females. Abbreviations. cancer (Ca); incidence (I); mortality (M).

*2.6. Study Risk-of-Bias Assessment*

A Study risk-of-bias (RoB) assessment was performed for the included observational studies based on the National Toxicology Program Office of Health Assessment and Translation (NTP OHAT) Risk of Bias Rating Tool for Human and Animal Studies [27]. This tool approaches all the different major types of study biases (i.e., selection bias, confounding bias, performance bias, attrition/exclusion bias, detection bias, selective reporting bias and other sources of bias) through 11 questions (Supplementary Table S2). Each question is classified either as a "key item" or "other applicable item" with the following response options: "--" (definitely high RoB), "-" (probably high RoB), "+" (probably low RoB) and "++" (definitely low RoB). Each study was graded as having a low RoB (Tier 1), moderate RoB (Tier 2) or high RoB (Tier 3), depending on the overall RoB assessment (Table 1 and Supplementary Table S2), with higher tiers indicating better internal validity. For the observational studies (cohort, case–control, cross-sectional), six study biases were evaluated based on seven questions (Supplementary Table S2). Tier 1 (T1) studies are studies with a definitely or probably low RoB, with all key items rated as "++" or "+" and having most other applicable items reported as "++" or "+". Tier 3 (T3) studies are studies with a definitely or probably high RoB, with all three key items rated as "--" or "-" and having most other applicable items answered as "--" or "-". Tier 2 (T2) studies are studies with a moderate RoB, classified as studies that met neither the criteria of the first nor the third tiers.

Across all human observational studies, we classified exposure (Q8), outcome (Q9), and confounding (Q4) assessments as "key items" (Supplementary Table S2), as these are the most frequently included elements for human observational studies [27].

*2.7. Data Synthesis*

The data synthesis follows a narrative presentation, discussing at first the established knowledge retrieved from the reviews and meta-analyses that included studies even before 2012. Then, we discuss the novel observational studies and how their findings add to the existing associations.

## 3. Results

*3.1. Study Selection Process*

The literature yielded 2009 records (Figure 1). After removing the duplicates, we obtained 1282 research papers and 362 reviews and meta-analyses. After the title and abstract review, 85 research papers and 10 reviews and meta-analyses were assessed for eligibility. Of these, 69 full-text articles and 3 reviews and meta-analyses were excluded (i.e., irrelevant outcomes, not estimating risk for incidence or mortality, case reports/case series, animal studies, studies in children, sample size < 50 participants, no exposure group, no control or comparison group, no Cr(VI) exposure, overlapping studies). In total, 16 observational studies (3 ecological, 2 pooled case–control, 2 case–control, and 9 retrospective cohorts) and 7 reviews and meta-analyses met the inclusion criteria for our systematic review.

*3.2. Study Risk-of-Bias (RoB) Assessment Results*

With regards to the RoB assessments for the observational human studies, we found four studies with a low RoB, nine with a moderate RoB and three with a high RoB (Table 1 and Supplementary Table S3).

*3.3. Gastrointestinal Cancers*

A total of 5 meta-analyses [29–33], 1 systematic review [8] and 16 observational studies [11–26] were published during the period of 2012–2023, inspecting the impact of Cr(VI) exposure on gastrointestinal cancer incidence and mortality. The most frequent types of gastrointestinal cancers addressed were gastric and hepatocellular cancer, reported in twenty and fourteen studies, respectively (Table 3).

### 3.3.1. Oral Cancer

One meta-analysis and one systematic review included oral cancer in their results, as shown in Table 3. The meta-SIR from the 16 studies included in the meta-analysis for oral cancer was significant and equal to 1.3 (95%CI [1.11, 1.54]), but the meta-SMR from 17 studies was non-significant (meta-SMR = 0.91; 95%CI [0.75, 1.10]) [30]. The systematic review's authors mentioned that the available evidence of the association between Cr(VI) exposure and oral cancer was insufficient, with some evidence indicating it may increase the risk of oral cancer [8].

The incidence of oral cancer was investigated in one study involving male cement workers [18]. The SIR for oral cancer was non-significant in the workers exposed to Cr(VI) compared with the national population. Oral cancer mortality was reported in nine studies, as presented in Table 3. In one cohort, the authors found a significant SMR = 2.03 (95%CI = 1.53, 2.63) for occupational exposure to Cr(VI) in men and oral cancer [17]. On the other hand, six cohorts and two ecological studies did not reveal a link between oral cancer and Cr(VI) exposure (Tables 1 and 3).

### 3.3.2. Esophageal Cancer

Esophageal cancer was investigated in one systematic review and one meta-analysis (Table 3). In their meta-analysis of 47 different cohort studies in workers with occupational exposure to Cr(VI), Deng et al. (2019) found a (marginally) significantly higher incidence (meta-SIR = 1.05; 95%CI [1.00, 1.11]), even though the mortality rate was insignificantly different (meta-SMR = 0.97; 95%CI [0.92, 1.01]) for digestive system cancer (esophageal, gastric, pancreatic, colon, rectum, hepatobiliary system and intestinal cancer) [30]. The respective meta-SIR and meta-SMR were not significant in the sub-group analysis for the cases of esophageal cancer. In another systematic review, den Braver-Sewradj et al. (2021) concluded that "There are no or no convincing indications that Cr(VI) can cause esophageal cancer in humans" [8].

The incidence of esophageal cancer was examined in three studies (Tables 1 and 3). In one retrospective cohort with more than 2 million workers, the risk for esophageal cancer was significantly higher in workers in industries manufacturing plastics and synthetic resins (HR = 2.27; 95%CI [1.02, 5.07]) compared to workers in the rest the manufacturing industries. However, the risk for esophageal cancer did not differ for the rest of the plastics and rubber manufacturing occupations [11]. In the remaining two studies, esophageal cancer was evaluated in workers in cement and manufacturing industries [18,22]. An association between Cr(VI) exposure and esophageal cancer was not detected. We found eight studies assessing esophageal cancer mortality (Tables 1 and 3). Cr(VI) exposure was primarily evaluated in electroplating, chromate production and printing industry workers from Italy, Spain, the USA, Taiwan and Russia. In one ecological cancer mortality study, 21,187 topsoil samples were collected across Spain to measure their levels of chromium [24]. Mortality due to cancer of the esophagus was significantly associated with higher chromium topsoil levels, but only in men. The remaining seven studies reported no significant impact of Cr(VI) exposure on esophageal cancer mortality (Table 3).

### 3.3.3. Gastric Cancer

Five meta-analyses and one systematic review were identified, with controversial results regarding the potential link between Cr(VI) exposure and gastric carcinogenesis (Table 3). The systematic review indicated that the existing evidence is limited and conflicting; however, a causal association between Cr(VI) exposure and gastric cancer was speculated [8]. One meta-analysis of 56 studies showed that Cr(VI) exposure is correlated with a higher risk of stomach cancer (meta-SIR = 1.17; 95%CI [1.09, 1.27]) and associated mortality (meta-SMR = 1.39; 95%CI [1.28, 1.51]) [32]. Deng et al. (2019) observed similar results for gastric cancer incidence (14 studies; meta-SIR = 1.20; 95%CI [1.08, 1.32]) but not for mortality (33 studies; meta-SMR = 0.93; 95%CI [0.78, 1.09]) [30]. In two meta-analyses that included cement workers ($n_1$ = 9 and $n_2$ = 26 studies included), no association was

with gastric cancer incidence or mortality [31,33]. Another meta-analysis including 44 observational studies found no significant overall risk for gastric cancer incidence or mortality (meta-RR = 1.08; 95%CI [0.96, 1.21]) [29].

Gastric cancer incidence and mortality were reported in four and ten out of the fourteen observational studies, respectively (Table 3). In one retrospective cohort involving cement industry workers, the risk of incidence was increased in the group with high dust exposure (SIR = 2.18; 95%CI [1.19, 3.65]) [18]. Similar results were recorded in a pooled case–control study using data from 11 different studies, with workers exposed to Cr(VI) having a higher odds of developing gastric cancer (OR = 1.51; 95%CI [1.30, 1.76]) [21]. In another retrospective cohort with more than 2 million workers, some job-specific subgroups in plastics and rubber manufacturing had a significantly higher hazard ratio for gastric cancer incidence [11]. On the other hand, one case–control study involving workers in various manufacturing industry categories (i.e., leather tanning, printing industries, etc.) did not highlight any significant relationship between occupational exposure to Cr(VI) and gastric cancer incidence [22]. Regarding gastric cancer mortality, in one ecological study, higher topsoil levels of chromium were significantly associated with higher mortality due to upper gastrointestinal tract cancer in the female population [24]. Another ecological study indicated a significant correlation between the distance from industrial production facilities for cement, lime and plaster and gastric cancer mortality in the male population [26]. Excluding these two high RoB ecological studies, all the other published papers indicate that exposure to Cr(VI) is not related to gastric cancer mortality (Tables 1 and 3).

### 3.3.4. Small Intestinal Cancer

Small intestinal cancer was studied in only one meta-analysis, with a non-significant meta-SMR = 0.98; 95%CI [0.81, 1.18] [30]. The insufficiency of published papers regarding the effects of Cr(VI) exposure on small intestinal cancer is also discussed by den Braver et al. (2021) in their systematic review [8].

Cancer of the small intestine was studied in only two published papers (Tables 1 and 3). Lifetime exposure to welding fumes was associated with an increased risk for some rare cancers, such as cancer of the small intestine (OR = 2.3; 95%CI [1.17, 4.50]), in one case–control study [20]. A prospective cohort of workers at a chromate production plant did not find any significant risk for gastrointestinal cancer mortality, including cancer of the small intestine [14].

### 3.3.5. Colorectal Cancer

In their systematic review, den Braver et al. (2021) did not find any convincing evidence that Cr(VI) was associated with colorectal cancer in humans [8]. Two meta-analyses studying colorectal cancer and exposure to Cr(VI) included all the published papers since 1980 (Tables 2 and 3). The first one included all studies involving cement workers during the period of 1980–2013 (n = 26) [33]. This group demonstrated an increased meta-SIR = 1.38 (3 studies; 95%CI [1.02, 1.88]) for colorectal cancer incidence but a non-significant meta-SMR = 1.05 (4 studies; 95%CI [0.79, 1.40]). As for the second study, digestive system cancer incidence (esophageal, stomach, pancreatic, colon, rectum, hepatobiliary system) was marginally affected by exposure to Cr(VI) (meta-SIR = 1.05; 95%CI [1.00, 1.11]), but this finding was not significant for cancer mortality [30]. When colorectal cancer was analyzed separately, the subjects exposed to Cr(VI) had no significantly increased risk for colon or rectum cancer incidence and mortality.

Eleven observational human studies investigated the impact of Cr(VI) exposure on colorectal cancer incidence (*n* = 2) and mortality (*n* = 9) (Tables 1 and 3). One case–control study found that printing industry workers were more likely to have colon cancer (OR = 1.37; 95%CI [1.15, 1.64]) [22]. A significantly higher SIR for rectal cancer was also observed in a retrospective cohort study of 1324 male cement industry workers (SIR = 3.05; 95%CI [1.32, 6.02]) [18]. Regarding cancer mortality, García-Pérez and colleagues (2015) found that residents in the vicinity of industries producing cement, lime and plaster had a signifi-

cantly higher risk of colorectal cancer mortality (RR = 1.08; 95%CI [1.03–1.13]) [26]. One retrospective cohort also showed that occupational exposure to Cr(VI) may increase the risk of rectal cancer mortality [16]. Although some studies have pointed out that exposure to Cr(VI) may have a causal effect on colorectal cancer mortality, the vast majority resulted in a non-significant effect on mortality rates (Table 3).

### 3.3.6. Hepatocellular Cancer

A single meta-analysis evaluated the impact of occupational exposure to Cr(VI) on hepatocellular cancer incidence and mortality (Table 3). According to this study, the meta-SIR calculated based on 4 studies was 0.92 (95%CI [0.76, 1.1]), and the meta-SMR based on 16 studies was 0.91 (95%CI [0.79, 1.04]) [30]. These results were in agreement with the conclusion of a recent review, suggesting that the available evidence cannot support an association between hepatocellular cancer and occupational exposure to Cr(VI) in humans [8].

Hepatocellular cancer was the second most studied cancer (*n* = 14 studies), with incidence reports in four studies and mortality in eleven studies, respectively (Table 3). Regarding incidence, workers in leather tanning, leather products and fur workers were more likely to have hepatocellular cancer (OR = 2.36; 95%CI [1.15, 4.83]) in one case–control study in more than 67,000 workers across various occupation groups [22]. These findings were in accordance with a pooled case–control study, which reported higher odds for hepatocellular cancer in male workers exposed to chromium (OR = 2.03; 95%CI [1.04, 3.99]) [23]. On the contrary, in their retrospective cohort studies, Koh et al. and DeBono et al. were not able to identify a significantly higher risk for hepatocellular cancer in workers exposed to Cr(VI) [11,18]. Two out of the eleven studies with mortality as an outcome reported a significant effect of exposure to chromium on hepatocellular cancer mortality (one only in males), as presented in Table 3.

### 3.3.7. Pancreatic Cancer

Occupational exposure to Cr(VI) did not significantly affect the risk of pancreatic cancer incidence and mortality [30]. This meta-analysis identified 47 papers with occupational exposure to Cr(VI), of which 8 evaluated the risk of pancreatic cancer incidence (meta-SIR = 1.04; 95%CI [0.89, 1.23]) and 16 evaluated the risk of pancreatic cancer mortality (meta-SMR = 0.94; 95%CI [0.81, 1.08]). One systematic review also concluded that there is insufficient evidence to support a causal effect of Cr(VI) exposure on pancreatic cancer incidence and mortality [8].

As presented in Table 3, ten human observational studies examined the potential correlation of exposure to Cr(VI) with pancreatic cancer incidence (two studies) and mortality (eight studies). In their case–control study, Kaneko et al. (2020) found that workers exposed to chromium (leather tanning, leather products and fur workers) had higher odds of developing pancreatic cancer (OR = 2.36; 95%CI [1.15, 4.83]) [22]. In another retrospective cohort study that included 2.18 million workers with cancer incidence as an outcome, workers in job-specific subgroups with exposure to Cr(VI) did not have a significantly higher risk of developing pancreatic cancer [11]. Regarding mortality, in their retrospective cohort involving 127 workers manufacturing chromium thin-layer plating, Girardi et al. found a significantly higher SMR [15], which was not the case for the remaining seven studies evaluating mortality (Tables 1 and 3). These studies were evaluated as having low, moderate and high RoB, and all of them had a larger sample size compared to Girardi's study.

### 3.3.8. Cancer of Gallbladder and Extrahepatic Bile Ducts

The results of the only meta-analysis that investigated gallbladder and extrahepatic bile duct cancer showed a non-significant meta-SIR and meta-SMR [30].

We found nine studies involving cancer of the gallbladder and extrahepatic bile ducts, two with incidence as the main outcome and seven with mortality as the main outcome (Table 3). Table 1 shows the characteristics of each study. Studies of low, moderate and high

RoB were included, whereas the study design varied from retrospective and prospective cohorts to case–control and ecological studies. The participants were mostly workers in occupations with high exposure to Cr(VI) (i.e., electroplating, chromate production, chromium thin-layer plating, cement industry, welding, shipbreaking) from different countries (Italy, the USA, Taiwan, Korea, China, Spain, Denmark, Sweden, Latvia, France, Germany). One ecological study also investigated the relationship between topsoil levels of chromium and all types of gastrointestinal cancer mortalities. None of the above studies reported any significant association between exposure to Cr(VI) and cancer of the gallbladder and extrahepatic bile ducts (Table 3).

*3.4. Benign Gastrointestinal Diseases*

We searched for all studies evaluating any association between Cr(VI) exposure and the most common gastrointestinal diseases (i.e., gastritis, gastroesophageal reflux disease, ulcer, irritable bowel syndrome, hemorrhoids, Crohn, ulcerative colitis, constipation, gastrointestinal bleeding, diverticulitis, celiac disease, gallstones, cholelithiasis, cirrhosis). We found one human observational study, as presented in Table 3. In this retrospective cohort study including approximately 5000 shipbreaking workers, the authors reported a significantly higher risk of cirrhosis mortality, but only in male workers (SMR = 1.32; 95%CI [1.01, 1.68]) [17]. Although exposure to known carcinogens like lead chromate, chromium and cadmium are commonly observed in shipbreaking workplaces, the above Tier 2 study is highly biased, as it does not assess other potential confounders or adjust for other exposure to heavy metals other than Cr(VI) [2]. In their systematic review, Hessel et al. (2021) explored any potential correlations between occupational exposure to Cr(VI) and non-cancer health effects [28]. The authors stated that occupational exposure to Cr(VI) might increase the risk of developing nasal septum, chronic lung diseases, skin ulcers, and allergic contact dermatitis, but there were no clear indications that Cr(VI) was associated with benign gastrointestinal diseases in humans.

**4. Discussion**

In our study, we reviewed the existing knowledge regarding exposure to Cr(VI) and the risk of developing gastrointestinal cancer and benign diseases. Some evidence indicates that exposure to Cr(VI) may increase the risk of developing gastric cancer, including the incidence outcome of two meta-analyses and three observational studies. However, little or conflicting evidence was found regarding mortality (2/10 observational studies and 1/5 meta-analyses reporting significant results). Exposure to Cr(VI) may be also associated with colorectal cancer incidence, as reported in one out of the two observational studies and one out of the two meta-analyses, but these results are not in accordance with the remaining studies. A higher colorectal cancer mortality was evident in only two out of nine of the observational studies and in neither of the two meta-analyses (Table 3). There was little or contradicting evidence that exposure to Cr(VI) may impact the incidence of oral cancer (one meta-analysis with significant results), and no evidence for mortality (one observational study reporting significant results only in men). There was no or inconclusive evidence that exposure to Cr(VI) may increase the risk of developing hepatocellular cancer (2/4 observational studies, 0/1 meta-analysis with significant results), esophageal cancer (1/3 observational studies, 0/1 meta-analysis with significant results), gallbladder and extrahepatic bile ducts cancer (0 studies with significant results) and pancreatic cancer (1/2 observational studies, 0/1 meta-analysis with significant results). The evidence was even less limited regarding mortality, since most of the studies found no significant associations with exposure to Cr(VI) (Table 3). Very little or inconclusive evidence exists for small intestinal cancer, since it was investigated in only two observational studies and one meta-analysis.

Regarding benign gastrointestinal diseases, we found only one published paper in the last decade (2013–2023) correlating the risk of developing cirrhosis with occupational exposure to Cr(VI) [17]. No meta-analyses were published during the same period. Our

findings are in agreement with the existing literature, as in their recent systematic review, Hessel et al. did not find any evidence that exposure to Cr(VI) is associated with benign gastrointestinal disorders [28]. Despite the lack of recent evidence regarding the association between Cr(VI) exposure and non-malignant gastrointestinal diseases, some older studies have examined this correlation. In a NIOSH Health Hazard Evaluation of workers in the electroplating industry in the USA, 5 of the 11 workers reported symptoms of gastric pain, 2 developed duodenal ulcers, 1 had gastritis, 1 experienced gastric cramps, and 1 experienced frequent indigestion; however, there was no control group [34]. In another study of 97 worker exposed to Cr(VI), 10 reported ulcer formation, and 6 of them reported hypertrophic gastritis [35]. These findings agreed with another two studies of chromate production workers with regards to duodenal ulcers [36,37]. As for liver diseases, in their cohort of 4227 workers exposed to Cr(VI), Moulin et al. (1993) reported an elevated SMR = 1.74 (95%CI = 1.31, 2.26) for liver cirrhosis [38]. Although these studies reported some evidence for an increased risk of developing duodenal ulcer and liver cirrhosis, they are not reliable due to their poor methodology, very small sample size (in most cases), and lack of a control group.

A variety of different risk factors for gastric cancer exist, with the most important being H. pylori, obesity, smoking, red meat, alcohol and a low socioeconomic status [39,40]. Exposure to heavy metals such as Cr(VI) can trigger gastric carcinogenesis through a variety of different pathways [41]. They can disrupt the protective barrier of the stomach's lining by reducing mucosal thickness, mucus content, and basal acid output, harming E-cadherin function and triggering reactive oxygen species. Reactive oxygen species, when stimulated, damage the gastric mucosal and alter the DNA, potentially harming signal transduction and cell growth [41]. These changes can ultimately lead to the development of cancer as well as an enhancement of malignant tumors. Heavy metals also stop the repair of DNA lesions or cause insufficient damage repair. They can also create anomalies in other genes and increase interleukin-8 production, a pro-inflammatory chemokine that facilitates carcinogenesis [41]. Based on our results, Cr(VI) exposure is suspected to increase the risk of developing gastric cancer. Our findings are in partial agreement with what is already supported by most international agencies, implying that there is little but suggestive evidence that exposure to Cr(VI) may cause gastric cancer [2,6,7]. Our study strengthens the evidence of this relationship by including recently published studies, but it also illustrates literature gaps to be addressed in future studies.

NIOSH reported that oral, esophageal and hepatocellular cancers might be caused by Cr(VI) exposure, with little evidence supporting this connection [2]. The meta-analysis by Deng et al. (2019) estimated a meta-SIR = 1.30 (95%CI [1.11, 1.54]) for oral cancer [30], but most studies did not find any significant association with Cr(VI) exposure. Similarly, inconsistent findings were observed in our review for hepatocellular, esophageal and pancreatic cancer. Cr(VI) can cause damage in the gastrointestinal tract and liver through different pathways that promote tumorigenesis [42], but the existing evidence cannot support a causal effect for oral, esophageal and hepatocellular cancer.

Based on the available data, the international agencies have concluded that there is no evidence regarding the association between small intestinal, colorectal, gallbladder and extrahepatic bile duct cancer and Cr(VI) exposure [2,5–7]. In our review, an increased risk of developing colorectal cancer was evident in some studies, with SMRs and SIRs that were significant and above 1.0 [18,22,26,33]. However, these findings are not consistent with those of the remaining included studies. It should be noted that only a few studies have been published on small intestinal cancer [14,20,30]. Cr(VI) exposure was not significantly associated with gallbladder or extrahepatic bile duct cancer in any of the papers discussed in our results.

Only a few retrospective observational studies have been published to assess environmental exposure to Cr(VI), as most studies have examined occupational exposure [6]. One study from Spain reported a significant correlation between the topsoil concentration of chromium and upper gastrointestinal tract cancer in females [24], but another study

in China did not find any significant associations [25]. An ecological study in Greece reported a high SMR for primary hepatocellular cancer in the residents of an area with elevated Cr(VI) levels in the public drinking water supply compared to a standard Greek population [43]. The association between elevated chromium levels in drinking water and gastrointestinal cancers was not evident in one mortality study in Nebraska, USA [44]. Regarding non-malignant gastrointestinal diseases, extremely little evidence is available to support any causal relationship. Some cases of gastrointestinal effects have been reported after oral exposure to Cr(VI) [5].

Among all the included observational studies, we found four with a low RoB, nine with a moderate RoB and three with a high RoB. Most of the studies (all of the moderate and high RoB studies) did not account for important confounding and modifying variables (i.e., exposure to other heavy metals), which is also a major limitation for our study. Three out of the four studies with a low RoB presented adjusted results for some confounders due to their methodological approach (matched case–control analysis) [20–22], and the last one included some important confounders, such as demographic characteristics, calendar period and exposure to other heavy metals [13]. Additionally, in some studies, the source of the exposure was not clearly defined, as they included a variety of different occupations or vaguely evaluated environmental exposure [11,17–19,24–26]. Most of the studies with a low or moderate RoB did not account for a dose–response relationship, which is an additional issue that should be examined when possible. In their retrospective cohort study (low RoB), Sciannameo et al. (2018) tried to quantify the relationship between Cr(VI) exposure and gastrointestinal cancer mortality [13]. They examined a dose–response relationship between the exposure to each agent (i.e., Cr(VI)) and the number of occupational years that the subjects were exposed to each agent. The authors then estimated the cumulative dose of exposure to each agent, as reported "by summing, across the different work periods, the product of the duration and the intensity of exposure, represented by the average tanks metal concentration in each company". The authors could not find any significant dose–response relationship between Cr(VI) exposure and gastrointestinal cancer mortality [13]. In their case–control study (low RoB), Kendzia et al. (2022) also tried to analyze a dose–response relationship, defined as the duration of welding, in years [20]. A significant dose–response association was reported regarding the risk of developing small intestine cancer (OR = 1.2; 95%CI = 1.02, 1.43), indicating that a greater number of years of exposure to welding further increased the risk of developing small intestine cancer. Two of the high RoB studies also tried to assess a dose–response relationship based on the topsoil levels of chromium in different areas [24,25]. In their ecological study, Chen et al. (2015) could not establish any dose–response relationship between Cr(VI) exposure and gastrointestinal cancer mortality [25]. On the other hand, Núñez et al. (2016) found a dose–response relationship, indicating that higher exposure to Cr(VI), defined as a higher topsoil concentration of Cr, was associated with higher mortality rates due to upper gastrointestinal tract cancers in females [24].

Regarding the methodological framework of the included observational studies, most were cohorts without a control or comparison group. These studies estimated the mortality (and, in some cases, incidence) of gastrointestinal cancers only in subjects who were occupationally exposed to Cr(VI) for years, comparing it with the expected incidence and mortality rate of the general population (produced SMR, SIR) [12,14–19]. All of the low RoB studies and two of the moderate RoB studies were either matched case–control studies [20–23] or retrospective cohort studies [11,13] comparing the estimated incidence or mortality rates between exposed and unexposed subjects. These studies frequently adjusted for some confounders (i.e., demographic characteristics), producing ORs or HRs. The last three of the high RoB studies were ecological studies. They estimated the mortality rates in different areas for the whole population and compared them with the topsoil levels of chromium or the distance from an industrial facility, producing an RR [24–26]. Whereas these studies included the concept of a dose–response relationship with Cr(VI) exposure, we cannot be confident in their exposure characterization.

Two systematic reviews were included in our paper. One of them discussed the carcinogenic effects of occupational Cr(VI) exposure, and the other discussed non-cancer, including all studies between 2012 and 2018 [8,28]. Although the authors discussed this association in detail, they did not include all the available studies, they did not mention any dose–response relationship and they did not assess the environmental exposure of Cr(VI). All the five meta-analyses assessed the potential impact of occupational Cr(VI) exposure in gastrointestinal cancers (incidence and morality), while none included benign gastrointestinal diseases or discussed environmental exposure. Additionally, a major limitation that all of these studies mentioned is the lack of confounders in most of the epidemiological studies they included. Three assessed this correlation only for gastric cancer [29,31,32]. Of the five meta-analyses, two included epidemiological studies only with workers in cement production [31,33], which further limits further generalizability of their findings. To our knowledge, this is the first systematic review investigating the association between exposure to Cr(VI) (either environmental or occupational) and both gastrointestinal cancer and non-malignant gastrointestinal diseases. We aimed to enrich the established evidence with novel studies that were already published, including reviews and meta-analyses which had not been taken into account yet. We collected all the published research papers using a double-review methodological approach and evaluation of the risk of bias. First, we collected all the recently published papers during the last decade, and second, we gathered all the published systematic reviews and meta-analyses during the same period, including all papers published since 1980. A major limitation is the high risk of bias in many occupational studies, mainly due to their comparison of incidence and mortality with a reference population. Most of the studies lacked information on the socioeconomic status, smoking status and exposure to other heavy metals or carcinogens, all of which are potential confounders of the association between Cr(VI) exposure and gastrointestinal cancer. Statistical adjustment for some of these confounding variables was presented in only four of the studies, which limits the credibility of the final findings. On the other hand, all the systematic reviews and meta-analyses faced the same issue, as it is extremely difficult to measure some of these confounders (i.e., level of exposure, exposure to other carcinogens) in these types of studies. Another critical limitation we should consider is that these studies did not usually evaluate a dose-dependent relationship with exposure to Cr(VI), but instead presented an overall comparison of exposure versus non-exposure. This might lead to altered results, as certain health issues might be associated with higher exposure levels of Cr(VI).

## 5. Conclusions

Our systematic review provides an update of the existing knowledge regarding the association between exposure to Cr(VI) and increased mortality and incidence of gastrointestinal malignant and benign diseases. Some weak evidence supports the conclusion that exposure to Cr(VI) may increase gastric and colorectal cancer risks. Evidence for oral, esophagus and hepatocellular cancer is unclear, as we did not find any convincing evidence. A significantly larger gap regarding small intestinal cancer is present. Future studies should primarily focus on incidence and mortality, as our review found significant results mostly in incidence and less in mortality, especially when addressing the association between Cr(VI) exposure and small intestinal cancer. A dose-dependent correlation should also be examined whenever possible. Benign gastrointestinal diseases might also be related to high exposure to Cr(VI) (i.e., damage in the gastrointestinal tract and liver) [42], but remarkably few studies are currently available. Environmental pollution is worsening day-by-day. For example, while the attributable burden of air pollution has been increasing over the last decades, a decreasing trend have been observed regarding the burden attributed to all households globally [45]. Although the evidence is yet unclear regarding the association between Cr(VI) and the risk of developing gastrointestinal cancer, increased concentrations of environmental Cr or other heavy metals have been observed in some cases [46,47]. It is

crucial to effectively manage the rising concentrations of chromium in the environment, especially in water and soil, to mitigate public health risks.

**Supplementary Materials:** The following supporting information can be downloaded at: https://www.mdpi.com/article/10.3390/environments11010011/s1, Supplementary Table S1: Literature search syntax; Supplementary Table S2: Office of Health Assessment and Translation risk-of-bias tool; Supplementary Table S3: Risk-of-bias assessment for each study.

**Author Contributions:** Conceptualization, K.K., K.T. and A.L.; methodology, K.K.; search strategy, K.K., D.V.D. and A.L.; study selection, K.K. and D.V.D.; evaluation of risk of bias, K.K. and D.V.D.; data extraction, K.K.; writing—original draft preparation, K.K. and D.V.D.; writing—review and editing, K.T., T.P. and A.L.; supervision, K.T. All authors have read and agreed to the published version of the manuscript.

**Funding:** This research received no external funding.

**Data Availability Statement:** Data are contained within the article and Supplementary Materials.

**Conflicts of Interest:** The authors declare no conflicts of interest.

## Abbreviations

| | |
|---|---|
| ATSDR | Agency for Toxic Substances and Disease Registry |
| NIOSH | National Institute for Occupational Safety and Health |
| RoB | Risk-of-bias |
| Cr(VI) | Hexavalent chromium |
| RR | Relative risk |
| SMR | Standardized mortality ratio |
| SIR | Standardized incidence ratio |
| OR | Odds ratio |
| HR | Hazard ratio |
| ASR | Age-standardized incidence rate |

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
