# Peer review of "The Impact of Exposure to Hexavalent Chromium on the Incidence and Mortality of Oral and Gastrointestinal Cancers and Benign Diseases: A Systematic Review of Observational Studies, Reviews and Meta-Analyses"

_environments, doi:10.3390/environments11010011_

Round 1

Reviewer 1 Report

Comments and Suggestions for Authors
  1. General comment:

  2. The review provides a comprehensive overview of the existing literature on chromium exposure and its potential association with gastrointestinal cancer risk. The systematic approach to the literature search and inclusion criteria enhances the credibility of the findings. However, further clarification on the methodology's intricacies and limitations would strengthen the overall interpretation of the results.

  3.  
  4. Specific comments:

  5.  

    • 1. The presentation of the results is clear and organized, with a focus on the major gastrointestinal cancers associated with chromium exposure. It would be beneficial to include a brief discussion on the quality of the included observational studies and reviews, providing insights into the robustness of the evidence.
    • 2. The distinction between malignant and non-malignant gastrointestinal diseases is well-articulated, but additional details on the specific diseases within these categories could enhance the readers' understanding.
    •  
  6. Constructive feedback:

  7.  

      The review systematically explores the relationship between chromium exposure and gastrointestinal cancer risk. While evidence suggests a weak association with gastric and colorectal cancer, the study highlights existing controversies in the literature, particularly concerning oral, esophageal, and hepatocellular cancer.
  8.  
  9. The conclusion could be strengthened by providing a nuanced discussion of the potential confounding factors and biases present in the reviewed studies. Acknowledging these limitations would contribute to a more balanced interpretation of the evidence.
    •  
    • Consider discussing any variations in the methodologies of the included studies and how these differences might impact the overall assessment of chromium exposure and gastrointestinal cancer risk.
    •  
    • Including recommendations for future research directions, especially addressing the identified gap in small intestinal cancer studies, would add depth to the conclusion.
    •  
  10. Summary:

  11. The review emphasizes a critical gap in research on small intestinal cancer and non-malignant gastrointestinal issues. Overall, further studies with rigorous methodologies are warranted to better understand the nuanced relationship between chromium exposure and various gastrointestinal outcomes.

Author Response

We uploaded a file with the reviewer's answers.

Reviewer 2 Report

Comments and Suggestions for Authors

The study "Environments 2023" is a systematic review that investigates the link between exposure to hexavalent chromium (Cr(VI)) and gastrointestinal health, particularly focusing on cancer and benign diseases. It compiles and analyzes recent research, including reviews and meta-analyses, to update the existing knowledge in this field. The review finds some evidence suggesting a potential increase in the risk of gastric and colorectal cancers due to Cr(VI) exposure, but the evidence is less clear for other gastrointestinal cancers. A major challenge identified in the study is the high risk of bias in occupational studies, which often do not adequately assess the dose-dependent effects of Cr(VI). The review highlights the need for future research to focus on more detailed aspects like incidence, mortality, and dose-dependency. Additionally, it emphasizes the importance of managing environmental chromium concentrations to mitigate public health risks. Overall, "Environments 2023" underscores the necessity for more comprehensive and methodologically sound research to better understand the impact of Cr(VI) on gastrointestinal health.

-        Assessment of Dose-Response Relationship: Currently, many studies do not assess the dose-response relationship with Cr(VI) exposure, but rather make a general comparison between exposure and non-exposure. A more detailed evaluation of this relationship could provide more accurate and informative results​​.

-        Focus on Incidence and Mortality as Primary Outcomes: Future studies should focus on incidence and mortality as primary outcomes, since most studies reviewed report significant results on incidence and few on mortality​​.

-        Examination of Dose-Dependency Correlation: Whenever possible, examine the dose-dependency correlation to better understand the impact of Cr(VI) exposure at different levels​​.

-        Research on Benign Gastrointestinal Diseases: Expand research on the potential correlations between high-dose exposure to Cr(VI) and benign gastrointestinal diseases, as there are currently few studies available​​.

-        To improve the presentation of the study, cite works where air quality has been considered in high-impact areas of the world, such as: Impact of air pollution on global burden of disease in 2019 10.3390/pr9101719

-        Effective Management of Environmental Chromium Concentrations: It is crucial to effectively manage the rising concentrations of chromium in the environment, especially in water and soil, to mitigate public health risks​​.

-        Improvement of Data Quality and Variance: It is necessary to improve the quality and variance of the collected data, including a wider range of occupational and environmental scenarios.

-        Integration of Studies from Different Geographical Backgrounds: Integrate studies from different geographical backgrounds to better understand the effect of Cr(VI) exposure in various contexts.

-        In-depth Analysis of Different Forms of Gastrointestinal Cancer: Focus on a more in-depth analysis of the different forms of gastrointestinal cancer to determine the specific effect of Cr(VI) exposure.

-        Exploration of Alternative Research Methodologies: Explore alternative research methodologies to overcome the limitations of current observational and occupational studies, such as experimental studies or more sophisticated predictive models.

Comments on the Quality of English Language

Clarity and Conciseness: The document sometimes includes lengthy sentences that could be made more concise. For example, in the section discussing the limitations of the systematic review, the sentence structure could be simplified for better clarity. Instead of "One major limitation of our systematic review is that most occupational studies were of high RoB for important confounding and modifying variables as these studies compared the incidence and mortality with a reference population," it could be revised to "A major limitation is the high risk of bias in many occupational studies, mainly due to their comparison of incidence and mortality with a reference population"

Consistent Terminology: Ensure consistency in the use of specific terms throughout the document. For instance, if "Cr(VI)" is used to denote hexavalent chromium, this abbreviation should be consistently used throughout the text to avoid confusion.

Grammar and Syntax: Some sentences show grammatical inconsistencies which could be refined. For instance, in the conclusion section, the phrase "Future studies should focus on incidence and mortality as primary outcomes since most studies in our review reported significant results on incidence and few on mortality" can be improved to "Future studies should primarily focus on incidence and mortality, as our review found significant results mostly in incidence and less in mortality".

Use of Abbreviations and Acronyms: The document uses several abbreviations and acronyms. It’s important to ensure that each abbreviation or acronym is clearly defined upon its first use in the text to aid reader understanding. For example, when "RoB" (Risk of Bias) is first used, it should be defined to ensure clarity for all readers.

Author Response

We uploaded a file with the answers to the reviewer.

Round 2

Reviewer 1 Report

Comments and Suggestions for Authors

The provided text summarizes a study on chromium exposure and its potential impact on gastrointestinal health, specifically cancer risk. While evidence suggests a weak association with gastric and colorectal cancer, the summary lacks details on methodology and quality assessment. Further clarification on non-malignant diseases and potential limitations would strengthen the overall presentation.

Author Response

You can find our respones in the attached file.
